# Feasibility of Continuous Monitoring of Endoscopy Performance and Adverse Events: A Single-Center Experience

**DOI:** 10.3390/cancers15030725

**Published:** 2023-01-24

**Authors:** Stephan Zandanell, Sophie Gensluckner, Gernot Wolkersdoerfer, Frieder Berr, Christiane Dienhart, Antonia Gantschnigg, Franz Singhartinger, Andrej Wagner

**Affiliations:** 1Department of Internal Medicine I, University Clinics Salzburg, Paracelsus Medical University, Müllner Hauptstrasse 48, 5020 Salzburg, Austria; 2Department of Internal Medicine, Rotthalmünster Hospital, 94094 Rotthalmünster, Germany; 3Laboratory for Tumour Biology and Experimental Therapies (TREAT), Institute of Physiology and Pathophysiology, Paracelsus Medical University, 5020 Salzburg, Austria; 4Department of Surgery, University Clinics Salzburg, Paracelsus Medical University, 5020 Salzburg, Austria

**Keywords:** endoscopy, patient safety, adverse events, performance measures, quality assessment

## Abstract

**Simple Summary:**

The endoscopic diagnosis and treatment of (pre-)malignant lesions play a central role in gastrointestinal oncology. The procedures can be high-risk, and a substantial proportion of patients have multiple co-morbidities. While the quality and safety of gastrointestinal endoscopy could be improved through the continuous monitoring of adverse events and performance measures, such monitoring, although recommended by various professional societies, is not yet established in clinical practice. We began a prospective monitoring system at our endoscopy unit in 2018. This study aims to evaluate the quality of the documentation in order to validate the method via systematic review of the data collected at our center.

**Abstract:**

Background: We integrated a standardized questionnaire focusing on adverse events and performance measures in gastrointestinal endoscopy as a mandatory component of the electronical medical record. Methods: This retrospective study was conducted using prospectively collected data on quality parameters and adverse events (AEPM) for all diagnostic and therapeutic endoscopic procedures at our center between 2018 and 2020. Results: A total of 7532 consecutive endoscopic procedures were performed in 5035 patients. The proportion of high-risk examinations and high-risk patients was 20% and 23%, respectively. Severe adverse events (AEs, *n* = 21) occurred in 0.3% of procedures and significantly more often in patients with an ASA score > II (0.6%, *p* < 0.01). We observed no long-term morbidity after severe AEs. Mortality was 0.03% (*n* = 2). Following screening colonoscopy (*n* = 242), four endoscopists documented AEPM in more than 98% of the examinations. The cecal intubation rate was 97%, and the mean adenoma detection rate 60%. The quality of lavage was documented in 97% (rated as good in 70% and moderate in 24%). Conclusions: The risk of adverse events is significantly increased in patients with an ASA score > II, which should be considered when choosing treatment methods and precautionary measures. Continuous recording of AEPM can be effectively integrated into the clinical reporting process, enabling analysis of the data and feedback to be provided to endoscopists.

## 1. Introduction

The endoscopic diagnosis and treatment of (pre-)malignant lesions play a central role in gastrointestinal oncology. The evaluation of adverse events and performance measures (AEPM) is crucial for the continuous improvement and safety of medical care. Due to the invasiveness of the procedures, its role is of particular importance in gastrointestinal endoscopy. Severe adverse events (AEs) after interventional endoscopic procedures occur in up to 10% of cases [1,2,3]. Examination-related (therapeutic endoscopy and emergency examinations) and patient-related factors (American Society of Anesthesiologists classification (ASA) > II) have been described in the literature as independent risk factors for the occurrence of severe AEs [4]. However, the systematic and prospective recording of AEPM, as recommended by the current guidelines of professional societies [5,6,7], has not yet been established, although the proportion of invasive high-risk procedures and high-risk patients (ASA > II) [8] is particularly high in inpatient care.

We began a prospective monitoring system for AEPM at our endoscopy unit in 2018. For this purpose, we developed a questionnaire based on an electronic module as a mandatory part of the endoscopy report and the medical record, which is processed in three steps (Figure 1). This study aims to evaluate the quality of the documentation in order to validate the method via systematic review of the AEPM data collected at our center. Furthermore, we investigated the influence of patients’ risk profiles, their choice of sedation type, and the training level of four endoscopists on these parameters.

## 2. Materials and Methods

In the period between January 2018 and December 2020, the AEPM forms of all endoscopic procedures (*n* = 7532) of all consecutive patients treated at our endoscopy ward (*n* = 5035) were evaluated. Propofol (1%, 10 mg/mL) and other sedatives (e.g., midazolam, 1 mg/mL) were given under continuous monitoring of oxygen saturation and heart rate. In addition, intensive blood pressure monitoring (at least every 5 min) and continuous ECG recordings were performed in patients with ASA > II and during high-risk procedures. Sedation was administered either by the endoscopists or by qualified nurses. The endoscopy team always consisted of two nurses and one endoscopist. In patients with an ASA stage > II, sedation performed by an anesthesiologist or intubation anesthesia was usually organized in advance (>24 h). In the case of a drop in oxygen saturation, we increased the oxygen supply and improved mechanical airway protection. In the case of apnea or inadequate spontaneous breathing, we applied mask ventilation.

All endoscopists were informed in detail about the study in advance and gave their written informed consent. The Ethics Committee of Salzburg waived the issue of a statement due to the purely retrospective analysis of the data for quality assurance and AEs. For analysis of the data, four endoscopists (two experienced and two less experienced endoscopists) were selected from a total of nine physicians, who performed the majority of the examinations. Endoscopists were considered experienced when they had carried out more than 1000 prior therapeutic endoscopic interventions. Endoscopists were considered less experienced if they had previously performed less than 300 colonoscopies and 500 gastroscopies. All patients gave their written informed consent to the endoscopic procedures in advance.

The AEPM report is a mandatory part of all endoscopic and medical reports. Part 1 of the AEPM form is filled out electronically by the endoscopist during endoscopy; then, it is printed and forwarded to the ward. In this part, information is provided on:Performance measures for screening colonoscopy (patients who had undergone prior colonoscopies in the last 10 years, excluding the current examination, and patients with an age of less than 50 years were not included), including retraction time, the number of polyps, and the quality of lavage, assessed according to a modified Boston preparation scale (BBPS), in which poor and inadequate levels of the BBPS were combined into one level (“poor”).The type of endoscopy procedure, including the specification of therapeutic measures.The choice of sedatives.Risk stratification according to the ASA classification.The depth of sedation according to Richmond [9].Immediate AEs (respiratory, cardiovascular, and specific endoscopy-related AEs such as hemorrhage, aspiration, and perforation).

The selection of the parameters is based on the recommendations of the German Society for Digestive and Metabolic Diseases (DGVS, [7]).

Part 2 of the AEPM form is filled out manually by the attending physicians on the ward after discharge of the patient and forwarded to the endoscopists. Here, information is provided on late AEs (respiratory, cardiovascular, and endoscopy-related AEs such as bleeding, perforation, and pancreatitis) and on pain (according to the visual analog scale).

Part 3 of the AEPM form is reviewed and supplemented electronically by the endoscopists during routine revision of the medical report according to the associated histological results. In this part of the process, AEs that occurred were classified according to the classification system as listed below.

Adverse events were divided into immediate and late AEs (respiratory, cardiovascular, and endoscopy-related AEs such as bleeding, aspiration, perforation, pain, or pancreatitis) regardless of severity. The latter included all AEs that occurred after patient discharge from endoscopy up to 30 days after the examination. In addition, AEs were classified by severity into minor and major AEs according to the treatment required and the consequences for the patients, with major AEs being any event that resulted in the need for treatment with prolongation of hospital stay. This classification of AEs is principally based on the Clavien–Dindo classification [10,11]. AEs were classified as follows:No AEs.Minor AEs, e.g., temporary drop in blood pressure under sedation, pain, or nausea, without prolongation of hospital stay. Clinical measures: analgesics, antiemetics, antipyretics, or i.v. fluid administration/electrolyte correction.Major AEs with a need for medical treatment including endoscopic therapy, parenteral nutrition, or blood transfusions with prolongation of hospital stay of <48 h.Major AEs requiring surgical, endoscopic, or radiologic interventions with prolongation of hospital stay of > 48 h.Permanent morbidity, or clinical deterioration.Mortality.

For the evaluation, an additional distinction was made between sedation-related (usually a respiratory, cardiovascular, or allergic) and endoscopy-related AEs (bleeding, aspiration, perforation, pain, or pancreatitis). During the observation period, all AEPM forms were continuously controlled by the clinic’s medical office for completeness. In this study, the quality of documentation of all severe AEs (*n* = 21) and all endoscopy-related early and late AEs (*n* = 39) was evaluated after crosschecking the medical records.

For univariate nonparametric comparisons, the Mann–Whitney U test was used for nominally distributed variables when the cross-tabulations did not exceed 2 × 2. For cross-tabulations larger than 2 × 2, chi-square testing and Cramer’s V, as well as Fisher’s exact testing, were performed to examine associations between nominally distributed variables. Multiple binary logistic regression was used to examine the influence of factors on AE rates after univariate analysis. *p* < 0.05 was considered significant. SPSS (IBM SPSS Statistics, version 26) was used for all calculations.

## 3. Results

### 3.1. Patient Characteristics

During the observation period, 7532 consecutive endoscopic procedures were performed at our center in 5035 patients with a median age of 62 years. We performed 1718 procedures in patients (*n* = 1100) with an ASA score over II (median age: 67 years). Patient characteristics are summarized in Table 1.

### 3.2. Internal Validation

AEPM forms were electronically completed for all endoscopic procedures. The ASA stage was not documented in only seven cases (0.1%). Immediate AEs were not specified in eight cases (4% of all immediate AEs). All the AEPM forms of inpatients were returned by the wards to the medical office. In two cases, the ASA stage was not correctly assigned. In 131 cases (1.7%), late AEs were not specified. These cases and the documentation of all major AEs were evaluated after review of the corresponding medical records. Due to a systematic evaluation error, it was impossible to evaluate the information on the screening indication for all the colonoscopies of one single endoscopist. Retraction time was recorded in only 56% of screening colonoscopies. In 37 colonoscopies with polyps, and in 28 colonoscopies with adenomas (22% and 17% of all screening colonoscopies, respectively), the polyps were not correctly classified for the calculation of the adenoma detection rate (ADR).

### 3.3. Sedation, Procedures, Patient Risk, and Experience Level

In the vast majority of cases (92%), we used propofol mono-sedation. No sedation was given in individual cases (5% of all endoscopic procedures; mainly gastroscopy and sigmoidoscopy). We preferred midazolam mono-sedation in patients with allergies to peanuts, soy, or chicken egg protein (0.2%), existing hypotension (systolic RR < 100 mmHg, e.g., bleeding), or increased cardiovascular risk (e.g., severe pulmonary arterial hypertension, 0.4%). Endotracheal anesthesia was always performed by anesthesiologists (0.7%). Furthermore, our anesthesiologist team accomplished sedation or endotracheal intubation in 14 cases in the context of acute emergency (0.2%).

Twenty percent of all examinations were therapeutic. Therapeutic gastroscopy included variceal ligation, argon plasma coagulation (APC), polypectomy, endoscopic mucosal resection (EMR), dilatation, endoscopic submucosal dissection (ESD), and all cases of therapy for upper gastrointestinal bleeding such as APC, clipping, and percutaneous endoscopic gastrostomy (PEG) tube placement. Therapeutic colonoscopy included polypectomy, endoscopic mucosal resection (EMR), APC, dilatation, ESD, and all cases of therapy for gastrointestinal bleeding such as APC and clipping. The endoscopy team consisted of nine physicians, although the majority of endoscopic procedures were performed by three experienced endoscopists and one who was less experienced (*n* = 7045, 94%). The proportion of high-risk patients was particularly high in high-risk examinations (32%). High-risk procedures were solely performed by or under direct supervision of experienced endoscopists (Table 2).

### 3.4. Adverse Events

Minor AEs occurred in 2.8% and severe AEs in 0.3% of procedures. The most common AEs were sedation-related minor respiratory and cardiovascular events (2.3%). Bleeding at endoscopy, i.e., bleeding after biopsy or polypectomy, which could be successfully stopped immediately, e.g., via clipping, occurred in 0.4% of cases. Severe AEs were mainly endoscopy-related late AEs, e.g., bleeding after endoscopic resections in the colon and after papillotomy (0.4%). Patients with a higher risk profile (ASA > II) experienced sedation-related and endoscopy-related AEs more frequently (5.3% vs. 3.1%). Particularly sedation-related minor AEs and severe late AEs occurred more often in high-risk patients. There was no difference in endoscopy-related immediate AEs (Table 3).

As expected, most AEs occurred during high-risk procedures, particularly in colonoscopies with endoscopic resection (EMR and ESD) and at ERCP, which were associated with one-third of all major AEs (Appendix A).

### 3.5. Immediate AEs

In 69 cases, transient drops in saturation (peripheral oxygen saturation, SpO2 less than 90% for more than 1 min) occurred. In another 31 cases, the drop was more severe (<80%), and in 13 cases, SpO2 was less than 70%. We managed all of these cases by pausing sedating medication, increasing the nasally administered oxygen rate and jaw thrust maneuver, and applying a nasopharyngeal tube, if necessary. No additional late AEs occurred in any of these cases. Apnea was not documented in any case, and aspiration occurred six times. There was one documented case of aspiration pneumonia with a need for oxygen supply and antibiotic therapy.

Hypotension is a common side effect after propofol administration. Mild hypotension (RR < 90/60 mmHg for more than five minutes) was documented in 24 cases. Severe hypotension of less than 60/40 mmHg occurred in four cases. These AEs were all successfully treated by pausing the sedating medication and through i.v. fluid administration. In two cases, we administered theodrenaline/cafedrine. Bradycardia with <50 beats/minute occurred in 18 cases, and tachycardia with >140 beats/minute occurred in seven cases. These transient AEs did not require drug administration.

In one case, a perforation occurred in the colon during an ESD and subsequently underwent emergency surgery (partial colon resection). In another case, bleeding had already occurred intraprocedurally in the colon during an ESD, which could initially be stopped endoscopically. This patient was readmitted 10 days after discharge with acute lower gastrointestinal bleeding and underwent emergency surgical management (ileoascendostomy). Fortunately, the postoperative development was uncomplicated in these patients with ASA >II, without long-term morbidity (AE type D).

In two elderly patients (in both cases with a history of chronic heart failure, aged 98 and 99 years, respectively), cardiovascular arrests requiring resuscitation occurred during gastroscopy (acute GI bleed) and ERCP (cholangitis), respectively. Despite immediate successful resuscitation, the patients died a few days later.

In summary, 39 examination-related (including six major AEs, two of which were class D) and 169 sedation-related cases (including two major AEs of class C and two lethal AEs) were documented among immediate AEs (Table 3).

### 3.6. Late AEs

Overall, only a few late AEs were documented. The majority were major AEs of class C or D (*n* = 13), without long-term consequences. Minor AEs (*n* = 5) were fever or pain in the setting of mild pancreatitis after ERCP. The vast majority of examination-related late AEs were cases of delayed bleeding after polypectomy (*n* = 11, including one delayed bleeding after ESD, already mentioned above, which required surgical treatment, and two cases of delayed bleeding after endoscopic variceal ligation, requiring transfusion) and post-ERCP pancreatitis (*n* = 5, three of which were considered major AEs). Table 4 contains all the severe AEs and their outcomes in detail.

### 3.7. Sedation Regime and AEs

Sedation-related (immediate) AEs occurred in 2.3% of all sedations. High-risk patients suffered significantly more sedation-related minor and severe AEs (2.8% of all procedures in patients with ASA > II, *p* < 0.01). No severe sedation-related AEs were documented for mixed sedation with propofol and midazolam. In mono-sedation with midazolam in patients with allergies to propofol, no sedation-related AEs were documented at all. However, this was not the case in patients in whom midazolam was administered at a low dose (<3 mg) because of the increased risk profile. Here, minor AEs occurred in 12.5% of cases (Appendix A).

### 3.8. Performance Measures

Eleven percent of all diagnostic and therapeutic colonoscopies fulfilled the criteria for screening colonoscopies (*n* = 242). The mean adenoma detection rate (ADR) was 60%, including 19 adenomas larger than 2 cm and five invasive carcinomas. The mean age of the patients was 67 years (±13.3). The proportion of patients with ASA > II was 16%, which was significantly lower than the proportion of high-risk patients for all endoscopic examinations (23%). Endoscopist 4 documented the highest ADR, performing his first 194 colonoscopies during the training phase. Of note, endoscopists were unsuccessful in classifying polyps in 37 cases (15% of all resected polyps). The quality of bowel preparation was predominantly good (70%). The lavage was inadequate in only 3% of cases (Table 5).

### 3.9. Risk Factors

After the exclusion of variables in the univariate analysis, we identified several independent risk factors in the multiple logistic regression analysis. Patient-related factors were age (only for sedation-related AEs) and ASA stage > II (sedation- and endoscopy-related, severe immediate, and late AEs; Table 6).

Among the procedure-related factors, only ERCP was identified among therapeutic procedures as a significant risk factor for sedation-related minor and late major AEs. Among the sedation types, mixed sedation with propofol and midazolam was an independent risk factor for sedation-related minor AEs. Midazolam mono-sedation in hypotensive and higher-ASA-stage patients was also associated with increased risk for sedation-related AEs. No endoscopist-dependent clustering of AEs could be demonstrated (data not included in Table 6).

## 4. Discussion

In the observational period, we prospectively recorded AEs and performance measures in over 7500 endoscopic procedures at our center. To our knowledge, this is the first evaluation of a routine prospective recording of acute and late AEs of all endoscopic procedures obligatorily implemented in the medical record. There are a few studies with similar backgrounds in the literature. The “Austrian Benchmarking ERCP” program [12] has been collecting data on ERCP from numerous Austrian centers on a voluntary basis for more than 10 years. Therapeutic success and AE rates were evaluated with respect to individual examination numbers and center size. At the end of each calendar year, each center received individual results compared to the pooled benchmark data of all participating centers in strict confidence. The so-called Amberg Perforation Project integrated a prospective recording of AEs in endoscopy (mainly perforations) into the medical record [13]. A large multicenter prospective German study [4] has evaluated data on acute sedation-related AEs integrated into the endoscopy report. Recently, Nass et al. [14] reported on a Dutch nationwide registry for adverse events and performance measures in colonoscopy, covering more than 80% of all endoscopy services in the country (90% of the data from hospitals, and 10% from private endoscopy service institutions). Of note, the percentage of patients with an ASA score >III was only six percent, and no data on ADR and sedation-related adverse events were provided in this study.

The process flow and integration of electronic data collection for the AEPM monitoring did not require significant adjustments during the observational period. We evaluated the response rates and quality of the documentation after studying selected medical records and through internal validation. Despite shortcomings in the documentation of withdrawal time and in the classification of polyps during screening colonoscopies, the quality of the documentation was generally high. The only study that has examined a similar system for the documentation of immediate AEs in the endoscopy report [4] does not include information on documentation quality or internal validation. Due to methodological reasons, the authors did not cover late AEs in this large-scale multicenter study. Behrens and colleagues [4] identified ASA stage > II as a risk factor for the occurrence of sedation-related AEs in their large analysis. In our collective, patients with an increased risk profile (ASA > II) had a significantly higher risk for not only sedation-related but also endoscopy-related AEs.

An advantage of midazolam, if compared with propofol mono-sedation, is the lower rate of hypotension (meta-analysis in [15]) and cardiorespiratory AEs [16]. In the latter study of 9547 patients, emergency examinations and a higher dose of propofol were independent risk factors for AEs. Therefore, we used midazolam mono-sedation in patients with an increased risk for adverse cardiovascular events and hypotension. However, other meta-analyses could not show an increased rate of cardiovascular side effects in propofol mono-sedation [17,18,19]. Furthermore, we found no severe sedation-related AEs in cases with mixed sedation (propofol and midazolam). This observation is consistent with the results of a meta-analysis that compared propofol mono-sedation with other regimens [20].

Therapeutic endoscopy is inherently associated with significantly higher rates of AEs. Nass and colleagues demonstrated a significantly higher rate of bleeding and perforation events in therapeutic colonoscopy [14]. Bleeding was also the most common adverse event in our therapeutic procedures. However, ERCP was the only independent examination-related risk factor in multivariate analysis for sedation-related and endoscopy-related AEs. Of note, the rates of major adverse events after ERCP and other therapeutic and diagnostic endoscopies are comparable with data from the literature (Appendix A). We used a modification of the established surgical Clavien–Dindo classification for AEs [10,11]. Of note, Nass and colleagues introduced and recently validated a similar classification for endoscopic AEs derived from the Clavien–Dindo system in order to provide a comparison of performance between different endoscopy services and medical disciplines [21].

Sedation-related AEs occurred at a higher rate in our study, compared with the large prospective study with 368,206 endoscopic procedures by Behrens and colleagues [4]. However, the authors considered only incidents requiring intensive care with endotracheal intubation and resuscitation as major events. Moreover, the internal validation of documentation quality is not evident in their study with more than 30 participating centers. Severe sedation-related AEs are described in the literature in 0.01 to 0.9% of cases (Appendix A). In our evaluation, beyond the case of aspiration pneumonia, only two severe but fatal sedation-related AEs occurred in very elderly patients (98 and 99 years of age, respectively: 0.03%). Both cases were emergencies (hemorrhage and cholangitis, respectively). Sedation-related mortality in endoscopy is reported in the literature in up to 0.5% of cases [22,23]. We described a relatively high rate of sedation-related minor events (2.26% vs. 0.3%, [4]), which were mainly trivial (e.g., brief hypoxia). We considered these incidents adverse events due to formal reasons and to assess whether they could predict subsequent (even more severe) AEs, which was not the case.

The performance measures in upper and lower gastrointestinal endoscopy, which are propagated by the European Society of Gastrointestinal Endoscopy (ESGE) Quality Improvement Initiative [5,6], have a proven impact on clinical outcomes and quality of life.

ADR is an independent factor for colorectal cancer incidence and the most frequently used measure of quality of screening colonoscopy worldwide. In a large national evaluation in Poland [24], a significant increase in mortality correlated with an ADR of less than 20%. Nowadays, the “benchmark” of an ADR > 25% is not considered to be up to date. The endoscopes in our department are all equipped with HD resolution and narrow-band imaging. Actual studies including all modern means of neoplasia detection during screening colonoscopy show an improved ADR of between 48% and 70% [25,26]. In the present evaluation, the ADR is higher than in other studies [6]. However, the high ADR in our evaluation is a consequence of the inpatient setting and the pre-selection of patients (e.g., weight loss and anemia), and the results could not be compared with screening results from large national studies. In a multicenter Austrian cohort study with 128,969 patients, which also included inpatients (24%), a mean ADR of 22.7% could be documented [27]. Of note, the mean age of the patients (61 years) was significantly lower than in the present study (67 years). Other performance measures such as “documentation of the quality of bowel preparation” and cecal intubation rate (>90% required [6]) exceeded the benchmark requirements with 97% each. Unfortunately, only one endoscopist at our center documented the withdrawal time consistently.

A limitation of all studies based on the self-reporting of AEPM by endoscopy units is the potential underassessment of AEs. In our study, we tried to overcome this limitation through internal validation. Furthermore, all measured parameters at endoscopy (e.g., oxygen saturation and blood pressure) were independently documented as a part of the electronic nurses’ protocol and could be used for cross-checking against AEPM monitoring data.

Another weakness of the actual study is that AEPM monitoring in its present version covers only a limited selection of recognized performance measures. Facciorusso and colleagues [28] identified factors leading to the recurrence of advanced colorectal adenoma, such as the size and histological grading of resected adenomas. Our documentation system harbors detailed information, such as polyp size and histological grading, including information on the endoscopic and histologic in toto resection of polyps. We are currently evaluating these data, which will be part of future publications. Furthermore, we assess and validate endoscopic performance measures in our center through the combined evaluation of AEPM monitoring, process organization measures, and reviews of selected medical records. In the future, we plan to employ an optimized version of AEPM monitoring (e.g., including upper GI performance measures) and a scoring system for internal validation. Table 7 contains the current assessment and validation of performance measures including a proposal for a scoring system. This validation could be the subject of a future prospective evaluation that also addresses feedback and targeted interventions to minimize the risk of adverse events and to improve the quality of our endoscopy service.

## 5. Conclusions

The continuous recording of AEs and performance measures is feasible. Further improvement of the documentation process and user-friendliness, as well as automation in data analysis, could contribute to an increase in documentation quality and data accuracy. The advantage of this over retrospective analyses is the immediate availability of data, which allows for timely feedback and targeted interventions.

The risk for AEs of sedation and endoscopic interventions is significantly increased in patients with an ASA score > II and with a higher age, which should be taken into account when determining the indication and choice of sedation regimen.

The quality of screening colonoscopy under inpatient conditions at our center outperforms the ESGE benchmarks in terms of ADR and cecal intubation rate. Nevertheless, there is room for improvement, especially in the area of the documentation of retraction time and histological polyp type.

The present prospective recording method could contribute to an improvement in the quality of care in gastrointestinal endoscopy. In the future, AEPM monitoring could be introduced as an inpatient counterpart to the certification of screening colonoscopy, which is already successfully established in private practice in Austria [29].

## Figures and Tables

**Figure 1 cancers-15-00725-f001:**
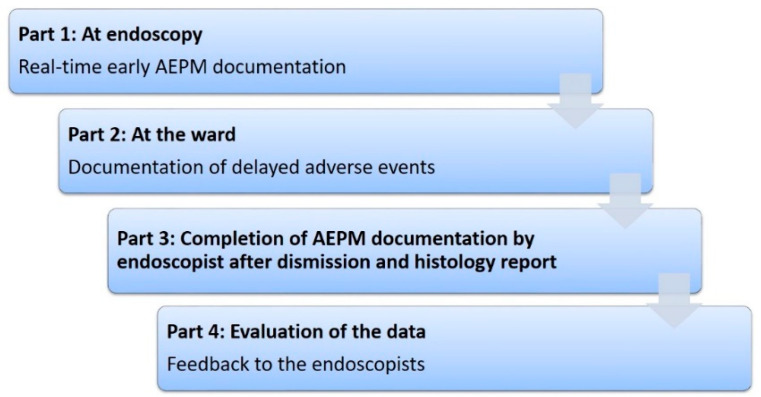
Process flow of AEPM monitoring. The AEPM form is a mandatory part of the endoscopic report and the medical record. It is processed in three steps: at endoscopy, on the ward after discharge of the patient, and, finally, by the endoscopist after evaluation of the patient’s clinical course.

**Table 1 cancers-15-00725-t001:** Patient characteristics.

2018–2020	Patients		Age		Endoscopic Interventions	
	*n*	%	Median (Range)	SD	*n*	%
Gender						
Male	2867	56.9	62 (18–94)	16.1	4330	57.5
Female	2168	43.1	62 (18–99)	18.3	3202	42.5
Risk category						
ASA I	915	18.2	48 (18–87)	16.1	1345	17.9
ASA II	3008	59.7	64 (18–97)	16.5	4462	59.2
ASA III	1084	21.5	66 (19–99)	13.9	1680	22.3
ASA IV	26	0.5	68 (19–91)	17.9	38	0.5
Total	5035	100	62 (18–99)	17.1	7532	100

Abbreviations: Mean: mean, SD: standard deviation, ASA: American Society of Anesthesiologists classification.

**Table 2 cancers-15-00725-t002:** Endoscopic procedures, endoscopists, and patient risk categories.

Type of Endoscopy	Total		Endoscopist		Patients ASA > II	
	*n*	%	Experienced (*n* = 3)	Less Experienced (*n* = 6)	*n*	%
Low-risk	6054	80.4	3818	2236	1234	20.4
Gastroscopy	3510	39.0	2157	1353	831	23.7
Colonoscopy	1791	19.9	1082	709	285	15.9
Sigmoidoscopy	291	3.2	173	118	60	20.6
Diagnostic EUS	400	4.4	382	18	40	10.0
PEG tube exchange	62	0.7	24	38	18	29.0
High-risk	1478	19.6	1166	312	479	32.4
Ther. gastroscopy	614	6.8	401	213	309	50.3
Ther. colonoscopy	405	4.5	313	92	108	26.7
EUS with FNA	67	0.7	65	2	9	13.4
ERCP	377	4.2	372	5	50	13.3
DBE	15	0.2	15	0	3	20.0
Total	7532	100.0	4984	2548	1713	22.7

Abbreviations: ASA: American Society of Anesthesiologists classification, EUS: endosonography, PEG: percutaneous endoscopic gastrostomy, Ther.: therapeutic, FNA: fine-needle aspiration, ERCP: endoscopic retrograde cholangiopancreatography, DBE: double-balloon enteroscopy.

**Table 3 cancers-15-00725-t003:** Adverse events of all endoscopic procedures, proportion of patients with ASA > II, and significant differences (univariate analysis).

Adverse Events	Total		Patients ASA > II		
	*n*	% ^1^	*n*	%	*p*-Value
Minor AEs	208	2.76	81	4.7	
Immediate AEs	203	2.27	81	4.3	
Sedation-related	170	2.26	73	4.3	0.001
Respiratory	118	1.57	47	2.7	0.001
Saturation-drop	114	1.51	47	2.7	
Aspiration	6	0.08	0	0.0	
Apnea	0	0.00	0	0.0	
Laryngeal edema	1	0.01	0	0.0	
Cardiovascular	52	0.69	26	1.5	0.001
Hypotension	28	0.37	13	0.8	
Tachycardia	7	0.09	7	0.4	
Bradycardia	18	0.24	5	0.3	
Hypertension	0	0.00	1	0.1	
Endoscopy-related	33	0.44	8	0.5	
Bleeding	32	0.42	8	0.5	
Epistaxis	1	0.01	0	0.0	
Late AEs	5	0.07	0	0.0	
Mild pancreatitis	2	0.03	0	0.0	
Post-bleeding	1	0.01	0	0.0	
Fever	2	0.03	0	0.0	
Major AEs	21	0.28	10	0.6	
Immediate AEs	9	0.12	4	0.2	
Sedation-related	3	0.04	2	0.1	
Aspiration	1	0.01	0	0.0	
Shock/reanimation	2	0.03	2	0.1	
Endoscopy-related	6	0.08	2	0.1	
Bleeding	1	0.01	1	0.1	
Perforation	4	0.05	1	0.1	
Surgery after perforation	1	0.01	0	0.0	
Late AEs	12	0.16	6	0.4	0.044
Pancreatitis	3	0.04	1	0.1	
Delayed bleeding	9	0.12	5	0.3	
Total	229	3.1	91	5.3	

Abbreviations: ASA: American Society of Anesthesiologists classification. ^1^ Related to *n* = 7532 total endoscopies; *p* < 0.05 considered significant, MWU test.

**Table 4 cancers-15-00725-t004:** Major AEs, risk profiles, and outcomes.

Adverse Events	*n*	%	ASA	Age Disease ^1^		Gender	AEC	Outcome	Type of Endoscopy
Immediate	9	42.9							
Sedation-rel.	3	14.3							
Aspiration	1	4.8	I					Pneumonia, RI	Ther. ERCP
Shock	2	9.5	III				F	Exitus letalis	Ther. gastroscopy
			III				F	Exitus letalis	Ther. ERCP
Endoscopy-rel.	6	28.6							
Bleeding	1	4.8	III	85	n.a.	f	D	Surgery, RI	Colonoscopy ESD
Perforation	5	23.8	II	70		m	D	Surgery, RI	Colonoscopy ESD
			II	79		f	C	Clip, RI	Ther. ERCP
			III	84	Hf	m	C	Clip, RI	Ther. ERCP
			II	65		m	C	Clip, RI	Colonoscopy EMR
			II	67		m	C	Clip, RI	Colonoscopy ESD
Late	12	57.1							
Delayed bleeding	9	42.9	III	80	n.a.	m	C	e.h., RI	Diagn. colonoscopy
			III	45	n.a.	f	D	e.h., RI	Diagn. colonoscopy
			II	52		m	D	e.h., RI	Ther. ERCP
			II	77		f	D	e.h., RI	Colonoscopy ESD
			III	73	Lf	m	D	e.h., TF, RI	Colonoscopy EMR
			III	43	Lf	m	D	e.h., RI	EVL
			II	69		m	C	e.h., RI	Colonoscopy EMR
			II	60		m	C	e.h., RI	ÖGD ESD
			III	55	Lf	m	D	e.h., TF, RI	EVL
Pancreatitis	3	14.3	II	57		m	D	Analgesia, RI	Diagn. ERCP
			I	56		f	C	Analgesia, RI	Ther. ERCP
			III	69	Lu	f	D	Analgesia, RI	EUS-FNA
Total	21	100							

Abbreviations: ASA: American Society of Anesthesiologists classification, AEC: adverse event classification, rel.: related, RI: restitutio ad integrum, ther.: therapeutic, ERCP: endoscopic retrograde cholangiopancreatography, End.: endoscopic, m: male, f: female, ESD: endoscopic submucosal dissection, EMR: endoscopic mucosal resection, e.h.: endoscopic hemostasis, Diagn.: diagnostic, EVL: endoscopic variceal ligation, TF: transfusion, EUS-FNA: endosonography with biopsy. ^1^ Underlying disease in cases with ASA > II (n.a.: not assessed, Hf: heart failure, Lf: liver failure, Lu: lung disease).

**Table 5 cancers-15-00725-t005:** Quality indicators of screening colonoscopy and quality of documentation.

Performance Measures for Surveillance Colonoscopy	Total		End. No. 1 (Experienced)		End. No. 3 (Experienced)		End. No. 4 (Inexperienced)	
	*n*	%	*n*	%	*n*	%	*n*	%
Colonoscopies total	2176	100.0	734	100.0	538	100.0	194	100.0
Ther. Colonoscopies total	361	16.6	119	16.2	152	28.3	60	30.9
Screening colonoscopies	242	11.1	71	9.7	137	25.5	31	16.0
Ther. colonoscopies	95	39.3	27	38.0	48	35.0	20	64.5
Colonoscopies with polyps	167	69.0	44	62.0	99	72.3	24	77.4
Colonoscopies with adenoma(s)	111	45.9	35	49.3	58	42.3	18	58.1
Colonoscopies with adenoma > 2 cm	19	7.9	2	2.8	17	12.4	0	0.0
Colonoscopies with carcinoma	5	2.1	2	2.8	3	2.2	0	0.0
Polyp not specified (adenoma)	37 (28)	15.3	3 (3)	4.2	30 (23)	21.9	4 (2)	12.9
Age ^1^	67	13.3	70	12.0	66	12.0	69	12.0
Male	130	53.7	40	56.3	73	53.3	17	54.8
Female	109	45.0	31	43.7	64	46.7	14	45.2
ASA > II	38	15.7	12	16.9	11	8.0	15	48.4
Quality of lavage good	169	69.8	50	70.4	95	69.3	24	77.4
Quality of lavage moderate	57	23.6	18	25.4	34	24.8	5	16.1
Quality of lavage bad	8	3.3	2	2.8	6	4.4	0	0.0
Quality of lavage documented	234	96.7	70	98.6	135	98.5	29	93.5
Cecal intubation	234	96.7	70	98.6	134	97.8	30	96.8
Documented withdrawal time	136	56.2	n.a.	n.a.	132	96.4	4	12.9
Adenoma detection rate	60.3		54.9		61.3		64.5	

Abbreviations: End.: endoscopist, ther.: therapeutic, ASA: classification of the American Society of Anesthesiologists. ^1^ Mean value, standard deviation.

**Table 6 cancers-15-00725-t006:** Selective multivariate logistic regression analysis. Factors and AEs.

Adverse Event	Sedation-Related							Endoscopy-Related Immediate							Endoscopy-Related Late				
	All					Major		All		Major					Major				
Variable	p univ.	OR	Lower OR 95% CI	Upper OR 95% CI	p multivar.	p univ.	p multivar.	p univ.	p multivar.	p univ.	OR	Lower OR 95% CI	Upper OR 95% CI	p multivar.	p univ.	OR	Lower OR 95% CI	Upper OR 95% CI	p multivar.
Age	******	**1.03**	**1.02**	**1.04**	******	n	n	*	n	*****	**0.02**	**0.75**	**1.04**	*****	**n**	**0.02**	**0.75**	**1.04**	*****
Gender	**	1.03	0.75	1.42	n	n	n	n	n	*	0.37	0.75	1.42	n	n	1.03	0.34	3.27	n
ASA > II	******	**2.52**	**1.77**	**3.61**	******	n	n	n	n	**n**	**20.87**	**1.77**	**3.61**	*****	*** ^1^**	**2.52**	**0.69**	**12.02**	******
OGIT diagnostic	**	6.89	0.34	138.27	n	n	n	*	n	**	1.01	0.34	138.27	n	n	6.89	0.00	-	n
UGIT diagnostic	n	2.36	0.11	49.20	n	n	n	n	n	n	0.00	0.11	49.20	n	n	2.36	0.00	-	n
OGIT bleeding	n	1.25	0.11	1.43	n	n	n	**	n	n	43.15	0.39	1.27	n	n	1.14	0.00	-	n
UGIT bleeding	*	0.56	0.39	1.27	n	n	n	n	n	n	12.82	0.63	2.04	n	n	1.12	0.11	2.14	n
Emergency ^2^	n	1.14	0.64	2.04	n	n	n	n	n	n	6.49	0.08	0.70	n	n	0.71	0.00	-	n
OGIT therapeutic	**	7.40	0.36	152.33	n	n	n	**	n	**	0.00	0.36	152.33	n	n	7.40	0.00	-	n
UGIT therapeutic	n	6.36	0.29	137.17	n	n	n	n	n	n	0.00	0.29	137.17	n	n	6.36	0.00	-	n
ERCP	**n**	**3.29**	**1.80**	**6.03**	******	n	n	n	n	**	17 × 10^4^	1.80	6.03	n	**n**	**3.31**	**0.21**	**4.57**	******
Propofol Mono	**	1.26	0.60	2.64	n	n	n	**	n	**	33 × 10^5^	0.60	2.64	n	**	1.26	0.00	-	n
Mixed sedation	******	**6.47**	**2.64**	**15.85**	******	**	n	**	n	**	0.77	2.64	15.85	n	n	6.47	0.00	-	n

Abbreviations: univ.: *p*-value in univariate analysis, OR: odds ratio, *p* multivar.: *p*-value in multivariate analysis, **: *p* < 0.01 in univariate analysis, *: *p* < 0.05 in univariate analysis, significant values are highlighted (bold), n: not significant, ASA: American Society of Anesthesiologists classification, OGIT: upper gastrointestinal tract, UGIT: lower gastrointestinal tract, Anesth.: anesthesia, ERCP: endoscopic retrograde cholangiopancreatography, Mid.: midazolam. ^1^ Significant only in the MWU test, and not in the chi-square test. ^2^ With anesthesia.

**Table 7 cancers-15-00725-t007:** Performance measures, assessment and validation.

Type of Endoscopy	Performance Measures [5,6]		Assessment		Internal Validation
		AEPM Monitoring	Process Organization ^1^	Medical Record ^2^	Scoring System ^3^
**Diagnostic esophago-gastroduodenoscopy**	6 h sobriety (2 h not drinking)		x		
	Documented examination duration of at least 7 min ^4^		x		
	Photo documentation of anatomical structures and relevant findings ^5^			x	10
	Use of standardized terminology		x e	x	10
	Simethicone/Sab (and possibly ACC) for preparation		x		
Screening colonoscopy	Documentation of the quality of bowel preparation	x			5
	Time window for a total colonoscopy of at least 30 min.		x		
	Cecal intubation rate >90% incl. photo documentation	x		x	5
	Adenoma detection rate (ADR) of >25%.	x			20
	Documented withdrawal time of at least 6 min (better 10 min, “target”)	x			10
	Adequate resection technique	(x)		x	10
	Adequate photo documentation of all pathological findings			x	10
	Use of standardized terminology		x	x	10
	Documentation quality of AEPM monitoring	x			10

^1^ Organizational guidelines of the clinic, standardized processes, validation through external audits; ^2^ Critical review of random endoscopy reports (*n* = 480) in the context of supervision as a part of an endoscopy training concept; ^3^ Proposal; ^4^ The beginning and end of each examination are recorded in the nursing documentation; ^5^ incl. Papilla vateri and all pathological findings (at least 10 standard images)

## Data Availability

Not applicable.

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
