# Peer review of "Feasibility of Continuous Monitoring of Endoscopy Performance and Adverse Events: A Single-Center Experience"

_cancers, 2023, doi:10.3390/cancers15030725_

Round 1

Reviewer 1 Report

The topic is of interest. I have the following comments:

1) Why did the authors use the Clavien-Dindo classification that is a surgical classifications of AEs instead of the validated ASGE lexicon? Otherwise, the authors should use the newer AGREE classification (PMID: 34890695 ) that is derived from the Clavien-Dindo.

2) The authors should comment how the quality of endoscopy reflects on the outcomes, for example in terms of adenoma recurrence (cite the recent paper PMID: 27005802)

3) English grammar should be improved. I recommend the authors to have their manuscript revised by a native speaker

Author Response

REVIEWER 1:

The topic is of interest. I have the following comments:

1) Why did the authors use the Clavien-Dindo classification that is a surgical classifications of AEs instead of the validated ASGE lexicon? Otherwise, the authors should use the newer AGREE classification (PMID: 34890695 ) that is derived from the Clavien-Dindo.

Thank you for this comment. We missed to be precise in this point. The Clavien-Dindo classification is not used in endoscopy, mainly because of the differences in postprocedural care (e.g., in outpatient care). Nass and colleagues 1 decided to base their classification system on a modified Clavien-Dindo system in order to “enable comparison of performance between … disciplines, such as GI endoscopy and surgery or interventional radiology”. We missed to clarify, that we used a modification of the Clavien-Dindo classification, too. Of note, the recently published AGREE classification is almost identical with our modified Clavien-Dindo classification from 2018. We have discussed these aspects in more detail in the revised version of the manuscript (“material and methods”, lines 128-129, 134, 137, 140, “discussion”, lines 49-53).

2) The authors should comment how the quality of endoscopy reflects on the outcomes, for example in terms of adenoma recurrence (cite the recent paper PMID: 27005802)

Thank you for the important comment. The quality of endoscopy is crucial for the patients´ clinical outcome. This fact was our main motivation, to implement AEPM documentation in 2018. Facciorusso and colleagues 2 identified factors leading to recurrence of advanced colorectal adenoma, such as size and histological grading of resected adenomas. Adenoma recurrence after endoscopic resection is a central issue, mainly addressed in post-polypectomy surveillance guidelines. In the actual study we focused on the established key performance measures, such as ADR and cecal intubation rate. Furthermore, our documentation system provides more detailed information, such as polyp size and histological grading, including information on endoscopic and histologic in-toto resection of polyps. We are currently evaluating this data, which will be part of future publications.

These issues have now been described in more detail and are now provided in the manuscript (“discussion”, lines 65-67, 93-98).

3) English grammar should be improved. I recommend the authors to have their manuscript revised by a native speaker

The manuscript has been checked by a native English-speaking colleague. Accordingly, we have modified and amended the manuscript.

  1. Nass K.J., Zwager L.W., van der Vlugt M., Dekker E., Bossuyt P.M.M., Ravindran S., Thomas-Gibson S., Fockens P. Novel classification for adverse events in GI endoscopy: The AGREE classification. Gastrointest. Endosc. 2022;95:1078–1085.e8. doi: 10.1016/j.gie.2021.11.038.
  2. Facciorusso, M. Di Maso, G. Serviddio, et al., Factors associated with recurrence of advanced colorectal adenoma after endoscopic resection. Clin Gastroenterol Hepatol, 14 (2016), pp. 1148-1154 e4

Reviewer 2 Report

In the present study, Zandanell and colleagues aimed at evaluate the quality of the documentation collected at the endoscopy unit in order to validate the method by systematical review of the AEPM data. Furthermore, they investigate the influence of patients’ risk profile, the choice of sedation type, and the training level of the endoscopists on these parameters.

The topic is very interesting and new. The paper is well designed and clear.

However I have some comments:

- In the "internal validation" part is written: ASA stage was not documented in only two cases. Immediate AEs were not specified in 8 cases (4% of all immediate AEs). All AEPM forms of inpatients were returned by the wards to the medical office. ASA stage was not recorded in only seven cases (0.1%), and immediate AEs were not specified in eight cases (0.01%). In this sentence there are some contradictions. Please check.

- Are we sure that a transient drops in saturation (peripheral oxygen saturation, SpO2, <90% and >80% for less than 1 minute) should be considered an adverse events?

Author Response

REVIEWER 2:

In the present study, Zandanell and colleagues aimed at evaluate the quality of the documentation collected at the endoscopy unit in order to validate the method by systematical review of the AEPM data. Furthermore, they investigate the influence of patients’ risk profile, the choice of sedation type, and the training level of the endoscopists on these parameters.

The topic is very interesting and new. The paper is well designed and clear.

However I have some comments:

- In the "internal validation" part is written: ASA stage was not documented in only two cases. Immediate AEs were not specified in 8 cases (4% of all immediate AEs). All AEPM forms of inpatients were returned by the wards to the medical office. ASA stage was not recorded in only seven cases (0.1%), and immediate AEs were not specified in eight cases (0.01%). In this sentence there are some contradictions. Please check.

Thank you for the comment. We apologize for the confusing phrase. We amended the corresponding section (“results”, lines 169-171)

- Are we sure that a transient drops in saturation (peripheral oxygen saturation, SpO2, <90% and >80% for less than 1 minute) should be considered an adverse events?

Thank you for this important remark. We apologize again for the mistake: it should be “SpO2 less than 90% for more than 1 minute.”  However, we fully agree that these minor events are relatively trivial (e.g., brief hypoxia, bleeding at polypectomy that is self-limited or easily treated endoscopically). We considered these incidents adverse events (AEs) in our statistics due to formal reasons and to assess whether they predict subsequent (even more severe) AEs. This fact may also account for the relatively high number of early AEs in our study.

We have discussed these aspects in more detail in the revised version of the manuscript (“discussion”, lines 63-67).

Round 2

Reviewer 1 Report

The revised manuscript is OK. Thank you!